# Production of Plant-Derived Japanese Encephalitis Virus Multi-Epitope Peptide in *Nicotiana benthamiana* and Immunological Response in Mice

**DOI:** 10.3390/ijms241411643

**Published:** 2023-07-19

**Authors:** Jae-Wan Jung, Pil-Gu Park, Won-Kyung Lee, Jun-Hye Shin, Mi-Hwa Jang, Eun-Hye Seo, Timothy An, Young Beom Kim, Myeong Hee Moon, Seuk-Keun Choi, Jee Sun Yun, Kee-Jong Hong, Seong-Ryong Kim

**Affiliations:** 1Department of Life Science, Sogang University, Seoul 04107, Republic of Koreami-hwa.jang@phytomab.com (M.-H.J.); 2PhytoMab Co., Seoul 04107, Republic of Korea; 3Department of Microbiology, Gachon University College of Medicine, Incheon 21936, Republic of Koreagmreo@naver.com (E.-H.S.); timothyan@naver.com (T.A.); 4BK21 Plus, Department of Cellular and Molecular Medicine, Konkuk University School of Medicine, Seoul 05029, Republic of Korea; 5Department of Chemistry, Yonsei University, Seoul 03722, Republic of Korea; bum5203@yonsei.ac.kr (Y.B.K.);; 6Eubiologics Co., Seoul 06026, Republic of Korea

**Keywords:** Japanese encephalitis virus (JEV), multi-epitope peptide (MEP), recombinant protein, transient expression, *Nicotiana benthamiana*

## Abstract

The current production of the Japanese encephalitis virus (JEV) vaccine is based on animal cells, where various risk factors for human health should be resolved. This study used a transient expression system to express the chimeric protein composed of antigenic epitopes from the JEV envelope (E) protein in *Nicotiana benthamiana*. JEV multi-epitope peptide (MEP) sequences fused with FLAG-tag or 6× His-tag at the C- or N-terminus for the purification were introduced into plant expression vectors and used for transient expression. Among the constructs, vector pSK480, which expresses MEP fused with a FLAG-tag at the C-terminus, showed the highest level of expression and yield in purification. Optimization of transient expression procedures further improved the target protein yield. The purified MEP protein was applied to an ICR mouse and successfully induced an antibody against JEV, which demonstrates the potential of the plant-produced JEV MEP as an alternative vaccine candidate.

## 1. Introduction

The Flavivirus genus belonging to the *Flaviviridae* family is composed of many clinically important human arboviruses such as Japanese encephalitis virus (JEV), West Nile virus, Dengue virus, yellow fever virus, Zika virus, tick-borne encephalitis virus, and St. Louis encephalitis viruses, causing encephalitis and haemorrhagic diseases [1]. Among these, JEV leads to an acute viral infectious disease of the central nervous system in part of Asia and the Torres Strait region of Australia [2]. JEV is well known to be found in animals such as pigs and carried to humans by a *Culex*-type mosquito that bites the JEV-infected animals [3]. Although most JEV infections are subclinical, it has been reported to be fatal in about 30% of people who experience symptoms [1,2].

The JEV genome is a positive-sense single-strand RNA consisting of three structural proteins of the capsid (C), the precursor to the membrane (prM) and E proteins, and seven non-structural proteins (NS1, NS2A, NS2B, NS3, NS4A, NS4B, and NS5) [4]. The C protein is involved in packaging viral genomes and forming nucleocapsids [5]. The prM protein acts as a chaperone for the folding and assembly of E proteins. The E protein plays an important role in virus adhesion and membrane fusion for entry into host cells [6]. The E protein is more conserved than other structural proteins and evokes the highest neutralizing antibody suggesting important epitopes for inducing protective human immunogenicity are contained in the E protein [7,8,9]. Furthermore, a multi-epitope peptide (MEP) containing six B-cell epitopes with two T-cell epitopes of E protein of strain SA14-14-2 expressed in *Escherichia coli* (*E. coli*) induced humoral and cellular immune responses, and provided protection against lethal JEV challenge in mice [10].

JEV vaccines currently used include inactivated mouse brain-derived vaccines, inactivated Vero cell-derived vaccines, live attenuated vaccines, and recombinant vaccines (World Health Organization (WHO), 2019), which have several issues to solve. The animal-based JEV vaccines have limitations including a lack of long-term immunity, the risk of allergic reactions due to the presence of brain-induced basic proteins or gelatine stabilizers, and high production costs [11]. In addition, there are concerns about the contamination of zoonotic viruses and endotoxins in animal cell-based antigen production [12]. A major disadvantage of attenuated vaccines is that secondary mutations can become toxic and cause disease, and people with weak immune systems are not recommended to receive the live vaccine [13]. Another limitation is that live attenuated vaccines typically require a cold chain to maintain effectiveness, making mass vaccination difficult in developing countries with limited facilities.

Plant-based vaccines have the theoretical advantages of being capable of high yields, rapid production, safety and low costs, and are unlikely to be contaminated with exotic mammalian pathogens [14]. Therefore, it has long been considered that plant-based platforms could be an alternative system to solve the issues of animal-based ones. Viral proteins such as SARS-CoV-2 spike protein could be expressed in the plant and virus-like particles formed [15]. Recently, Medicago Inc. has completed phase 3 clinical trials for a flu vaccine derived from *Nicotiana benthamiana*. The plant-based quadrivalent virus-like particle (QVLP) flu vaccine showed similar effects to the existing vaccine [16,17]. Moreover, the plant-based COVID-19 VLP vaccine (Covifenz^®^) produced by Medicago was authorized for use by Health Canada in February 2022, which is the first plant-based vaccine approved [18]. Plants are drawing attention as a new platform for producing protein drugs.

In this study, we hypothesized the production of JEV-MEP antigen using a plant system. Here we report the successful expression and purification of the JEV-MEP antigen from *N. benthamiana* and its ability to induce an immune response in mice (Appendix A).

## 2. Results

### 2.1. Vector Construction for N. benthamiana Expression System

To express JEV-MEP in plants, vectors were constructed using an epitope-based vaccine sequence [10]. As shown in Figure 1a, JEV RNA contains seven non-structural proteins and three structural proteins, E, C, prM. The JEV-MEP sequence consists of six B-cell epitopes (amino acid residues 75–92, 149–163, 258–285, 356–362, 373–399, and 397–403) and two T-cell epitopes (amino acid residues 60–68 and 436–445) in E protein (Figure 1b, Appendix A). Three different plant expression vectors, carrying codon-optimized JEV-MEP with 6× His-tag and that contain 33KD protein signal peptide or FLAG-tag at the N-terminus, were constructed into the pHREAC2 vector, a modified version of the pHREAC vector [19] containing the synthetic CPMV 5′ and 3′-UTR (Figure 1c). pSK474 and pSK475 were, respectively, designed to accumulate the target protein in either cytoplasm or apoplast. Additionally, pSK480 was designed to improve and facilitate the detection and purification of target proteins [20].

### 2.2. Expression of JEV-MEP

The *N. benthamiana* leaves infiltrated with *A. tumefaciens* carrying each vector were harvested at 4 day post-infiltration (dpi) and used for expression of JEV-MEP; it was confirmed and its levels of the three different constructs were compared on Western blot at 4 dpi (Figure 2a). The protein bands corresponding to 20 kDa in both pSK474 and pSK480, and 24 kDa in the pSK475 were observed, which means the expression of JEV-MEP in *N. benthamiana* by three individual constructs. The increased size of JEV-MEP in pSK475 might be due to the failure of SP cleavage or post-translational modifications during the secretion pathway to the apoplast. The signal from pSK480 was the strongest on the Western blot membrane, which showed about 10-fold higher intensity than that from pSK474 and pSK475. Therefore, pSK480 was chosen for JEV-MEP production in *N. benthamiana* and further experiments.

### 2.3. Optimization of JEV-MEP Expression in N. benthamiana

To optimize the JEV-MEP production level in *N. benthamiana* leaves, various conditions were investigated such as the dpi, the concentration of *A. tumefaciens* (OD_600_), and acetosyringone concentration in infiltration buffer. The expression level of JEV-MEP peaked at 4 dpi (Figure 2b), when the OD_600_ = 0.4 concentration of *A. tumefaciens* was infiltrated (Figure 2c) with 300 μM acetosyringone (Figure 2d). Notably, OD_600_ higher than 0.6 caused the necrosis of leaves and it might decrease the expression level (Figure 2c). It suggests the optimization of conditions of agroinfiltration can increase the expression level of JEV-MEP in *N. benthamiana*. The fixed optimal condition was used for further production of JEV-MEP. Additionally, both soluble and insoluble fractions from leaf extracts were loaded onto the Western blot membrane, and it showed the presence of JEV-MEP protein in the soluble fraction, indicating that the JEV-MEP is soluble (Figure 2e).

### 2.4. Purification of JEV-MEP

JEV MEP in pSK480 carries both FLAG-tag at N-terminus and 6× His-tag at C-terminus; therefore, two different methods were tested for the purification to determine the optimal purification strategy. Both purification methods could be used for the purification of JEV MEP, but the purified JEV MEP from the FLAG-tag purification method showed a higher amount of target-sized protein but a smaller amount of plant host protein than that of His-tag purification (Appendix A). Therefore, the FLAG-tag was used for the purification and further experiments. Purified JEV-MEP was visualized on the Coomassie-stained gel and Western blot membrane (Figure 3), and quantified by comparing band intensity with that of commercial recombinant JEV E protein produced in HEK293 cells using the Image J program. The measured amount of JEV-MEP in the crude extracts and purified fraction was 58 ng and 390 ng, respectively. These levels equate to overall yields of approximately 5.8 and 1.3 mg of JEV-MEP protein per Kg wet mass of infiltrated leaves. Bands of two, three, four, and five-fold higher than the monomer form of JEV-MEP were also observed on Western blot, respectively. This seems to be because of multimerization of JEV-MEP.

### 2.5. Induction of Protective Immunity against JEV by Vaccination with Purified JEV-MEP

To evaluate plant-derived JEV-MEP as vaccine material against JEV, an in vivo experiment using a mouse model was conducted. All mice used in the experiment did not show clinical symptoms (anorexia, dullness, humpback, etc.), suggesting that JEV-MEP does not cause side effects in mice. For serological analysis, sera obtained from mice which were immunized with 20 μg of purified JEV-MEP at an appointed timepoint were subjected to ELISA and PRNT analysis (Figure 4a). The results of ELISA analysis showed that JEV E-specific antibody is efficiently induced by JEV-MEP immunization (Figure 4b, Appendix A). Although statistical significance was not confirmed due to variation between individuals in IgM antibody titre (right panel in Figure 4b, lower panel in Appendix A), the average OD_450_ value of IgG in the vaccinated group was over 3.0, which is 10–20 times higher than that of the negative control group (left panel in Figure 4b, upper panel in Appendix A). The result indicates that the JEV-MEP antigen significantly increased the specific antibody titre against the JEV.

As the actual protective ability of vaccine substances is determined by the ability to induce neutralizing antibodies, not by simple binding antibodies, we also evaluated the neutralizing antibody in sera. As shown in Figure 4c and Appendix A, JEV incubated with sera which were obtained from JEV-MEP immunized mice showed significantly reduced plaque formation ability. These results suggest that plant-based JEV-MEP can efficiently induce protective immunity, as a vaccine substance against JEV infection.

## 3. Discussion

### 3.1. Expression and Multimerization of JEV-MEP

JEV E protein is associated with binding to cell receptors and membrane fusion during viral infections and has both B- and T-cell epitopes [21] that are required for epitope-based vaccines. JEV MEP protein, a kind of chimeric protein composed of multiple epitopes in JEV E protein, was successfully expressed in *N. benthamiana*. The 20 kDa-sized MEP was detected on Western blot; however, MEP protein expressed in leaves infiltrated with pSK475 showed a 24 kDa-sized band (Figure 2a) that is higher than expected. There are two possible reasons, the uncleaved signal peptide and post-translational modifications such as *N*-glycosylation. Cleavage of the signal peptides from secretory proteins is mediated by a signal peptidase [22] but the poor presentation of the processing site to signal peptidase can limit this process [23,24]. Glycosylation, which is one of the post-translational modifications of proteins, occurs at asparagine residues (N-linked) or serine, threonine residues (O-linked) [25]. One of each N- and O-glycosylation site of JEV-MEP was predicted with the program (NetOGlyc-4.0, NetNGlyc-1.0). Additions of glycans add to the size of protein in immunoblot. Therefore, the size shift may be due to one of these reasons or both. Optimizations of the conditions for transient expression such as the day for harvest after infiltration, and the concentration of *A. tumefaciens* and acetosyringone highly increased the expression level of JEV MEP protein in *N. benthamiana* (Figure 2b–e) [26] and the selection of purification improved the target protein yield. The purification yield of JEV-MEP target protein FLAG-tagged (pSK480) was about 22.4%. FLAG-tag is known to be necessary when trying to purify highly pure proteins at low yields when highly specific detection is necessary [20]. The association between the tagged structure and protein stability should be investigated more precisely in further studies. Although the purification yield and purity were improved by attaching the FLAG-tag, still a large amount of plant host proteins was contained in eluted fractions. The use of FC fragments for purification can increase the purity of the target protein to ~90% [27]. We tested the Fc fusion at the C-terminus of the JEV MEP protein (MEP-Fc), expression in *N. benthamiana*, and its purification (Appendix A). It highly increased the expression level and purity of eluted fractions (Appendix A) but most MEP-Fc protein was still in the flow-through. We tested eluted fractions to determine whether MEP-Fc can induce an immune response in the mouse but it showed negative results. The low binding efficiency and poor antigenicity of MEP-Fc protein seemed to be because of its aggregation, so we checked its native conditions using Western blot with reducing, non-reducing, boiling, or non-boiling (Appendix A), and it showed a very high level of aggregation. To dissolve these soluble aggregates, we tested purified MEP-Fc with the increased pH (pH 9.0), stirring overnight, and the treatment of dithiothreitol (DTT), and only DTT slightly increased the intensity of the band that is expected to the dimer form of MEP-Fc (Appendix A). In several studies, the aggregations of target protein occurred in the expression host or low pH conditions during the purification step caused, and it could be solubilized by using urea or non-ionic detergent such as Tween-20 [28]. We tested these for MEP-Fc protein, but could not observe the efficient dissociations of aggregates (Appendix A). It might be improved by the use of modified Fc fragments with minimized aggregation [29,30,31].

In contrast, MEP-FLAG induced an immune response in mice although we observed JEV-MEP multimerization during the expression and purification stages (Figure 3). This seems to be because epitopes of MEP-FLAG were exposed or the monomer form was the most abundant. Multimerization of MEP could be due to protein–protein interaction such as disulphide bonding. The first and seventh epitopes in JEV-MEP contain cysteine residues (Figure 1b), and cysteine residues can form disulphide bonds possibly causing the dimerization of protein monomers [32]. This can be tested in further works to define whether cysteine residues in the first and seventh epitopes cause aggregation, by removing or modifying epitopes or residues. It was also noted that plant-expressed proteins containing soluble molecules such as antibodies, VLPs, and HIV gp120 had a higher proportion of multimers or aggregates than CHO cells and yeast-expressed proteins [33,34], but the reason has not been investigated [35]. Controlling the cross-linking of plant-produced proteins could raise interest in the production of plant-based pharmaceuticals and biological products [36], so it is important to develop the methods for this and genetic engineering technologies such as CRISPR-Cas9 [37].

### 3.2. Immunogenicity of JEV-MEP

The recombinant JEV-MEP protein expressed in plants greatly induced JEV-specific antibody production in mice. Consistent with previous studies on seroconversion after infection or immunization with filovirus antigen [38,39,40], the serum IgM titre showed a relatively low level and peaked at the early timepoint (5 weeks after first immunization of antigen), and showed a tendency to decrease thereafter, while the serum IgG titre remained high throughout the observation period (Figure 4b, Appendix A). Induced antibodies successfully neutralized the infective JEV in PRNT assay. Approximately 20 of PRNT_50_ titre was observed in our assay as we used sera obtained at 6 weeks after immunization (Figure 4c, Appendix A), which is the timepoint of declining IgM titre; sera obtained at the precise timepoint are expected to have a higher neutralizing effect against the virus.

The plant-based production system is expected to secure a safer protein production platform than the animal cell-based production platform because the use of animal-derived ingredients that may cause animal pathogen contamination [41] is not required. Furthermore, the target protein, JEV MEP, is one of the antigen subunits, which is known as a safer version than live or attenuated for the vaccination. These advantages enable safer production of JEV vaccines, although there are concerns about the potential immunogenicity of plant-specific N-glycans such as β-1,2-xylose and α-1,3-fucose [42,43,44]. However, any clinical cases of these concerns were not reported [45,46].

Here, we report the recombinant JEV-MEP protein was successfully expressed in *N. benthamiana* and its immunogenic activity as a vaccine against JEV was confirmed using animal experiments. Although, in this study, we did not evaluate the protective ability of JEV-MEP against the death of mice caused by actual virus infection due to the limitation of space in animal BSL2 facility, further in vivo investigation is expected to more clearly demonstrate the excellent vaccine efficacy of the plant-based JEV-MEP. The results of this study suggest that a plant-based expression system can serve as an alternative for the production of effective and useful vaccine candidates against JEV infection.

## 4. Materials and Methods

### 4.1. Plasmid Construction for Expression of JEV-MEP

A sequence called JEV-MEP in which six B-cell epitopes and two T-cell epitopes from JEV strain SA14-14-2 linked with dipeptide of glycine (G) and serine (S) [10] with 6× His-tag at the end of C-terminus was used for vector construction. The nucleotide sequence of JEV-MEP was codon-optimized for the expression in *N. benthamiana*, and then commercially synthesized (GenScript, Piscataway, NJ, USA) and introduced into the plant expression vector pHREAC vector [19] (a kind gift from H. Peyret, John Innes Institute, Norwich, UK) to give plasmid pSK474 using *Bsa*I cleavage site. The N-termini of synthetic DNA was modified by PCR and cloned into the pHREAC vector using the *Bsa*I cleavage site. Briefly, pSK475 and pSK480 contain JEV-MEP with 6× His-tag carrying 33KD protein signal peptide at the N-terminus or FLAG-tag at the N-terminus, respectively.

### 4.2. E. coli and A. tumefaciens Transformation

The individual plasmids were transformed to *E. coli* (DH5α) by heat shock method and selected on the lysogeny broth (LB) agar plates containing 50 mg/L kanamycin. Colonies were picked for polymerase chain reaction (PCR) to confirm the recombinant clones. The positive clones that showed the expected sized band (500 bp) were reconfirmed by DNA sequencing after plasmid purification. The right clone was transformed into *A. tumefaciens* by the freeze–thaw method. Briefly described, purified vector DNA was added to *A. tumefaciens* (LBA4404) competent cells and left in liquid nitrogen for 5 min. After giving heat shock at 37 °C for 5 min, it was kept in ice for 2 min and then recovered in S.O.C medium at 28 °C for 2 h. The cells were spread on LB agar plates containing 50 mg/L kanamycin and 50 mg/L rifampicin, and then grown at 28 °C for 48 h. Confirmed colonies by colony PCR were used for agroinfiltration using *N. benthamiana*.

### 4.3. Transient Expression of JEV in N. benthamiana Leaves

*A. tumefaciens* carrying expression vectors inoculated in 5 mL LB media (50 mg/L kanamycin, 50 mg/L rifampicin) was incubated at 200 rpm at 28 °C for 48 h. The cells were collected and re-suspended with infiltration buffer (10 mM 2-N-morpholino-ethanesulfonic acid (MES), 100 mM MgCl_2_, 300 μM acetosyringone, pH5.5) to an OD_600_ of 0.2–0.8 at OD600. The wild-type *N. benthamiana* plants grown under 16 h light/8 h dark conditions at 25 °C for 4 weeks were used for agroinfiltration. The leaves were infiltrated with *A. tumefaciens* using a needleless syringe and then harvested at 3 to 6 dpi. The protein was extracted from leaves with phosphate-buffered saline (PBS, pH 7.4) buffer with cOmplete-EDTA Free™ (Roche Diagnostics GmbH, Mannhein, Germany). The extracts were clarified by centrifugation at 14,000× *g* at 4 °C for 10 min. The proteins in the supernatant and the pellet were analysed by sodium dodecyl sulphate–polyacrylamide gel electrophoresis (SDS-PAGE) and immunoblotting analysis.

### 4.4. Immunoblotting Assay

The extracted protein was analysed using SDS-PAGE and immunoblot under reducing conditions. Commercial recombinant JEV E protein produced in HEK293 cells (REC31688, The Native Antigen Company, Oxford, UK) was used as a positive control. Protein samples were separated on 12% acrylamide gel and the gel was stained with Instant Blue (Abcam, Cambridge, UK). For immunoblotting, proteins were transferred to a nitrocellulose membrane (Bio-Rad, Hercules, CA, USA). The membrane blocked with 5% skim milk at room temperature for 1 hour was incubated with 1:2500 dilution of rabbit polyclonal anti-JEV antibody (PA5-111964, Thermo Fisher Scientific, Waltham, MA, USA), followed by 1:5000 dilution of anti-rabbit IgG antibody (HRP) (GTX213110-01, GeneTex, Irvine, CA, USA). Chemiluminescence was detected using an ImageQuant LAS 500 (GE Healthcare, Chicago, IL, USA). The quantification of proteins was estimated by comparing the band intensity with the positive control using the Image J program (https://imagej.nih.gov/ij/download.html (accessed on 24 May 2021)).

### 4.5. Protein Purification

The clarified plant extract was filtered by syringe filter through a 0.45 µm syringe filter (Merck Millipore, Sartorius, Burlington, MA, USA) and used for purification using affinity resin for either 6× His or FLAG (Thermo Fisher Scientific, Waltham, MA, USA). After incubation of the sample with resins for 1 h at 4 °C, the resin was washed with wash buffer (100 mM phosphate buffer with 200 mM NaCl, pH 7.4 for FLAG column; 50 mM phosphate buffer with 300 mM NaCl and 50 mM Imidazole, pH 7.4 for Ni-NTA column). The recombinant protein was finally eluted with elution buffer (0.1 M glycine-HCl, pH 2.8 for FLAG column; 50 mM phosphate buffer with 300 mM NaCl and 300 mM imidazole, pH 7.4 for Ni-NTA column). The purified proteins were analysed using SDS-PAGE and immunoblotting. The total soluble protein in the plant extracts was estimated by using Bradford assay (Bio-Rad, Hercules, CA, USA) or Pierce™ BCA Protein Assay kit (Thermo Fisher Scientific, Waltham, MA, USA) to measure the antigen protein concentration.

### 4.6. Analysis of JEV-MEP Specific Antibody in the Mouse by ELISA

The verification of the antibody titre against the JEV in the serum was conducted. Three-week-old female ICR mice were immunized intramuscularly using 20 μg of purified JEV-MEP per mouse. All mice in the experiment group were boosted twice at an interval of 2 weeks. The negative control group was injected with 50 µL of PBS. Blood was collected from mice on day 0 and at 2, 4, 5, and 6 weeks, and the antibody titres in the sera were measured by ELISA. Mouse anti-JEV E protein IgG and IgM antibody titre was analysed using an ELISA kit (Alpha Diagnostic Int., San Antonio, TX, USA) as per the manufacturer’s protocol. Briefly, the collected mouse serum was diluted with dilution buffer provided at a 1:50 ratio in 100 μL and loaded into the JEV E protein-coated 96-well plate, and incubated at room temperature for 1 h. Anti-JEV E protein antiserum enclosed in the kit was used as positive control. The plate was washed and then reacted with 100 μL of an anti-mouse IgG-HRP or IgM-HRP conjugate as a secondary antibody, and incubated at room temperature for 30 min. The colour was developed by adding 100 μL of TMB substrates for 10 min. The reaction was terminated by adding 100 μL of stop solution. The optical density was measured at 450 nm using a microplate reader (Molecular Devices Corporation, Sunnyvale, CA, USA). All the animal experiments were approved by the Institutional Animal Care and Use Committee at Konkuk University (assurance number: KU21103) and carried out by the recommendations in the guide of the committee.

### 4.7. Titration of Neutralizing Antibody against JEV

Sera obtained at 6 weeks after the first immunization were diluted at a concentration of 1/10, 1/20, and 1/40, then incubated with the Nakayama strain of JEV for 2 h at 37 °C under 5% CO_2_. Then, the virus–sera mixture was administered to BHK-21 cells for 1 h, which were then incubated with culture media containing 1% low melting temperature agarose (Lonza, Rockland, ME, USA) for 120 h. After fixation of cells with 4% (*w*/*v*) formaldehyde solution (Biosesang, Seongnam, Korea), viral plaques were visualized with 1% (*w*/*v*) crystal violet solution (Sigma-Aldrich, Saint Louis, MO, USA). The PRNT_50_ titre was defined as the reciprocal of the serum dilution which reduced the number of plaques by 50% compared to the average plaque number of PBS control and was calculated through linear regression with data of 3 dilution points using Excel program (Microsoft, Redmond, WA, USA). All experiments with infectious viruses were approved by the Institutional Biosafety Committee of Gachon University (assurance number: GIBC-2022003) and conducted in the Biosafety Level 2 facility at Konkuk University.

### 4.8. Quantification and Statistical Analysis

The results of multiple experiments are presented as the mean ± standard error of the mean. For the analysis of ELISA results, statistical analysis was performed using a two-tailed Welch’s *t*-test, using Excel program (Microsoft, Redmond, WA, USA). A *p*-value < 0.05 was considered statistically significant.

## Figures and Tables

**Figure 1 ijms-24-11643-f001:**
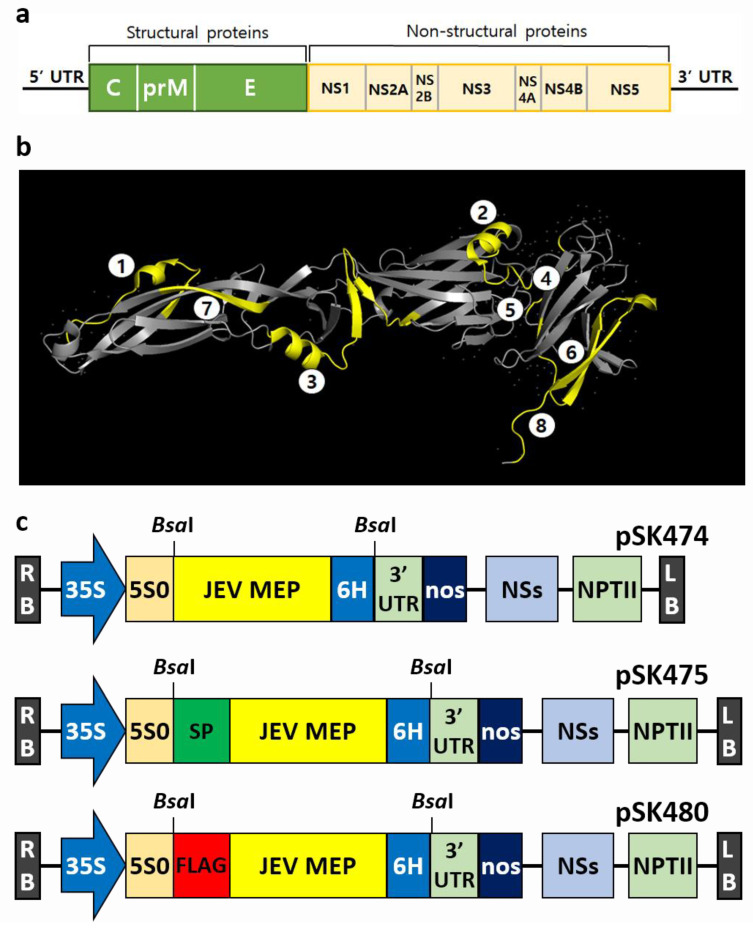
**Structure of JEV genome and MEP, and schematic representation of plant expression vectors.** (**a**) JEV genome organization. Structural proteins (C, prM, and E) and seven non-structural proteins (NS1, NS2a, NS2b, NS3, NS4a, NS4b, and NS5) are displayed. (**b**) Three-dimensional structure analysis of JEV E protein (PDB ID 3P54) drawn with PyMOL. Eight epitopes are represented in yellow. (**c**) Vectors for the transient expression. Each target gene was inserted in the pHREAC vector. Vectors carrying MEP fused with 6× His-tag at the C-terminus alone (upper, pSK474), and with additional signal peptide (middle, pSK475) or FLAG-tag at the N-terminus (lower, pSK480). LB, T-DNA left border; RB, T-DNA right border; 35S, 35S promoter of cauliflower mosaic virus; 5S0, synthetic 5′UTR; 6H, 6× His-tag; 3′UTR, 3′UTR of cowpea mosaic virus RNA-2; nos, NOS terminator; NSs, silencing suppressor from Tospovirus Tomato zonate; NPTII, neomycin phosphotransferase.

**Figure 2 ijms-24-11643-f002:**
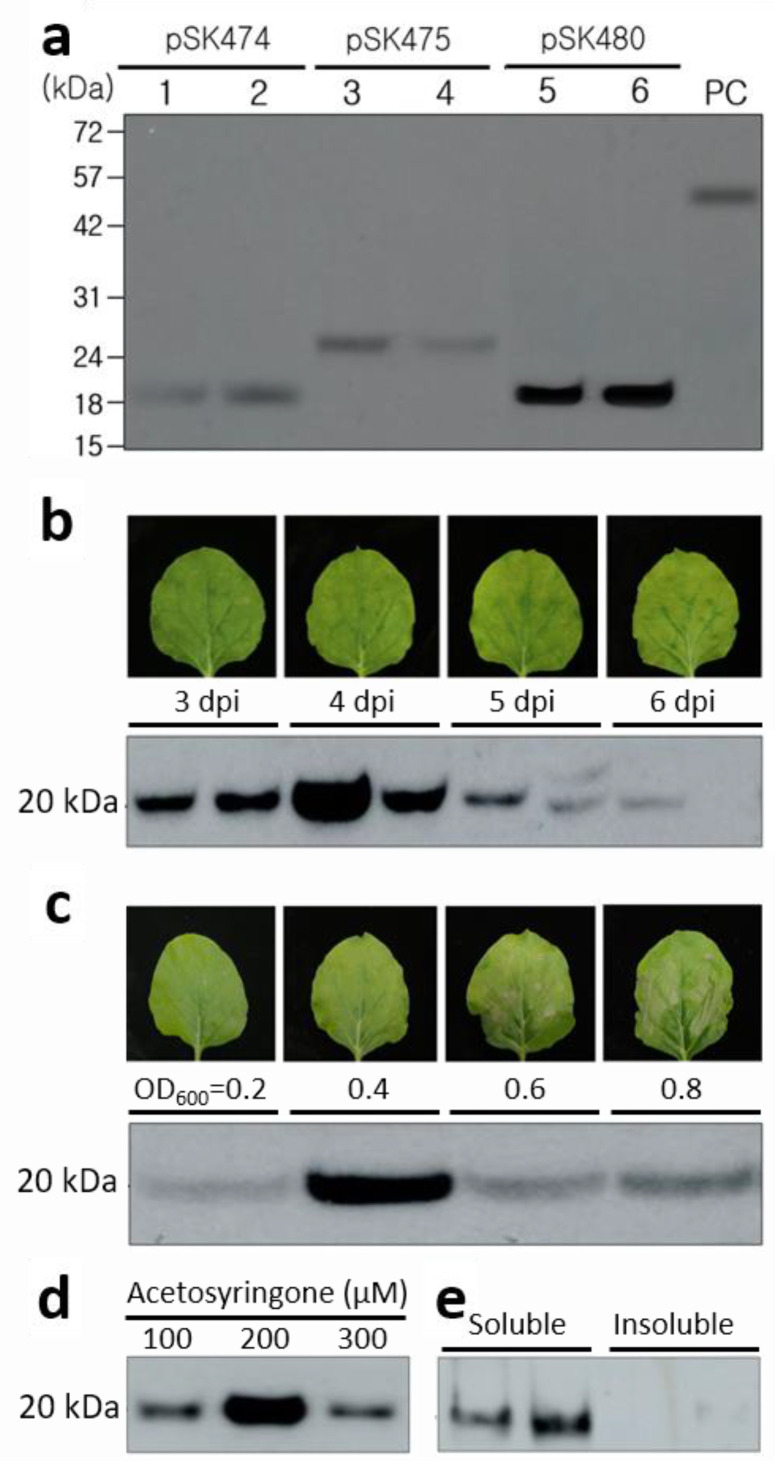
**Expression of JEV MEP and the optimization of conditions for agroinfiltration.** (**a**) Comparison of the expression level of JEV-MEP in the leaves infiltrated with three different vectors by immunoblotting. In all cases, expression was confirmed at 4 dpi. Lanes 1 and 2, pSK474; Lanes 3 and 4, pSK475; Lanes 5 and 6, pSK480; PC, twenty ng of recombinant JEV E protein (45 kDa) produced in HEK293 cells. Optimizations for the agroinfiltration of pSK480 by adjusting the day for leaf harvest (**b**), the concentration of *A. tumefaciens* (**c**), and the concentration of acetosyringone (**d**) to increase expression level. Images of leaves show their conditions. The presence of JEV MEP in the soluble part after protein extraction was observed using Western blot (**e**).

**Figure 3 ijms-24-11643-f003:**
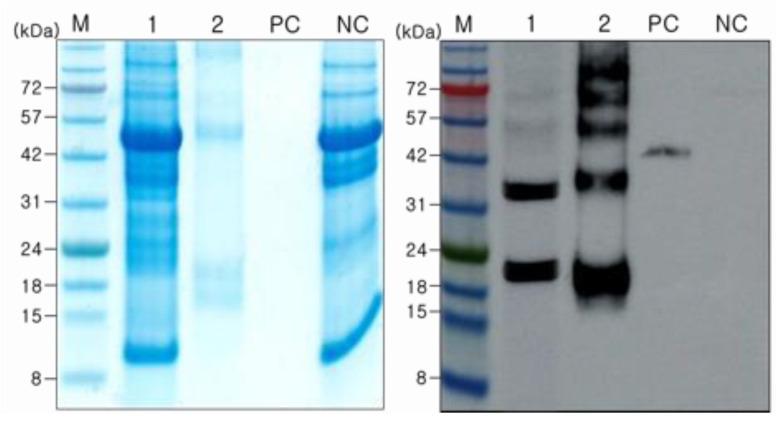
**Purified JEV-MEP by using FLAG-tag.** JEV MEP expressed in the leaves infiltrated with pSK480 was purified by using a FLAG-tag. The JEV MEP protein in the samples before and after purification was observed in SDS-PAGE (left) and immunoblot analysis using polyclonal anti-JEV antibody under reducing conditions (right). Lane 1, before purification; lane 2, purified JEV-MEP; PC, twenty ng of recombinant JEV E protein (45 kDa) produced in HEK293 cells; NC, non-infiltrated wild-type control.

**Figure 4 ijms-24-11643-f004:**
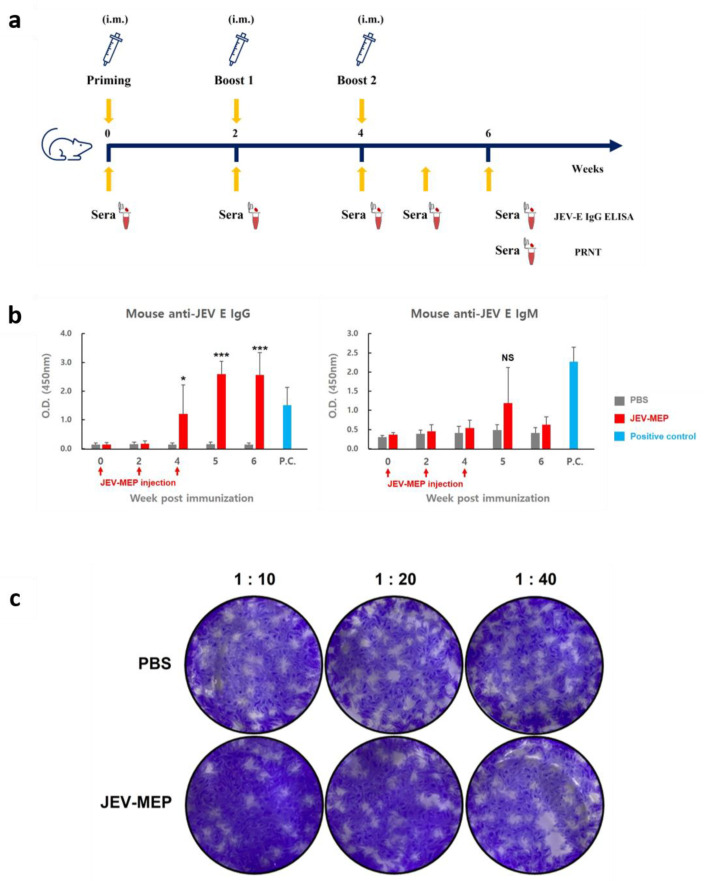
**In vivo analysis of plant-derived JEV-MEP vaccination.** (**a**) Schematic diagram of JEV-MEP vaccination and analysis schedule. Twenty µg of purified JEV-MEP was administered intramuscularly to three-week-old female ICR mice (N = 7) to induce protective immunity against JEV. The negative control group (N = 4) was injected with 50 µL of PBS. All mice were boosted twice at an interval of 2 weeks and sera obtained from each timepoint were subjected to ELISA and PRNT assay. (**b**) Serum antibody titre against JEV in JEV-MEP vaccinated mouse. Anti-JEV E-IgG antibody titre (left panel) and -IgM antibody titre (right panel) were measured by the indirect ELISA method. Anti-JEV E protein E antiserum was used as positive control (N = 3). (* *p* < 0.05, *** *p* < 0.001; two-tailed Welch’s *t*-test). See also Appendix A. (**c**) Neutralizing effect of serum purified from JEV-MEP vaccinated mice on JEV. Neutralizing activity was verified by numerical measuring of plaques induced by the JEV incubated with serum. Based on the plaque numbers matched with PBS control and each diluted serum, the PRNT_50_ titre is calculated as 18.5. See also Appendix A.

## Data Availability

The unedited images of the western blots shown in the figures are available in the Appendix A.

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
