# Peer review of "Production of Plant-Derived Japanese Encephalitis Virus Multi-Epitope Peptide in Nicotiana benthamiana and Immunological Response in Mice"

_ijms, 2023, doi:10.3390/ijms241411643_

Round 1

Reviewer 1 Report

In this manuscript, Jung has established a plant-based expression system to produce Japanese Encephalitis Virus (JEV) vaccine using the chimeric protein composed of antigenic epitopes from the JEV envelope protein. The authors have investigated using various vector constructs to express JEV multi-epitope peptide (MEP) sequences and compared the protein expression level and yield in purification. Importantly, in vivo experiments have demonstrated that the purified JEV-MEP can induce the immune system in mice vaccinated by the purified JEV-MEP. Overall, the data is consistent with the conclusions and the plant-based system can produce JEV-MEP which can potentially be used as vaccine material against JEV. However, some statistical analysis is missing in the manuscript and additional experiments are needed to further support the efficacy of the purified JEV-MEP in inducing immune response.

Specific comments:

1)    The title of the manuscript needs to be edited to align with the major contribution of this study which has established the plant-based JEV-MEP expression system which can boost immune response and therefore be used as vaccine to protect against JEV.

2)    Figure 2a, the raw image with marker labeled in the original blot needs to be shown.

3)    Figure 3, mass spectrometry needs to be performed on the purified protein with the target size to confirm the correct protein sequence. Since the purified protein doesn’t have high purity and there are multiple bands reacted with the antibody based on immunoblot analysis, other major bands should also be sent for mass spectrometry analysis.

4)    Figure 4a, more information is needed for the mice experiments including age, gender and how mice are monitored throughout the experiments. Mouse body weight before and after JEV-MEP vaccination needs to be monitored and shown.

5)    Figure 4b, at least 3 mice need to be included in control / PBS group for statistical analysis. In addition to showing the results from each individual mouse, summarized data with mean and standard deviation needs to be plotted between PBS and JEV-MEP groups. The results need to be compared using statistical analysis and p-value needs to be presented.

6)    Figure 4c, the raw data calculated for PRNT titer needs to be shown and statistical analysis with mean and standard deviation needs to be presented.

7)    Figure 4, the positive control group using standard vaccine material to immunize mice is missing.

The safety concerns of using the plant-based expression system of JEV-MEP as vaccine against JEV need to be discussed.

The English language needs some editing. 

Author Response

To reviewer,

We would like to thank you for your review and corrections. Regarding major and minor points, we corrected the manuscript and explained it here in red color.  

Specific comments:

1)    The title of the manuscript needs to be edited to align with the major contribution of this study which has established the plant-based JEV-MEP expression system which can boost immune response and therefore be used as vaccine to protect against JEV.

=> The title of the manuscript is now changed to “Production of Plant-derived Japanese Encephalitis Virus Multi-Epitope Peptide in Nicotiana benthamiana and Immunological Response in Mice”

2)    Figure 2a, the raw image with marker labeled in the original blot needs to be shown.

=> Figure 2a is now changed as your recommendation.

3)    Figure 3, mass spectrometry needs to be performed on the purified protein with the target size to confirm the correct protein sequence. Since the purified protein doesn’t have high purity and there are multiple bands reacted with the antibody based on immunoblot analysis, other major bands should also be sent for mass spectrometry analysis.

=> We expect those bands are multimeric forms of MEP-Flag because the sizes of those bands were about 40, 60, 80 and 100 kDa, which is multiple to monomer size (20 kDa). And also, this band was not detected from the negative control. Regarding multimerization of MEP protein, it is discussed in the Discussion part. We added more explanation about it in the Results, at the end of section 2.4. 

4)    Figure 4a, more information is needed for the mice experiments including age, gender and how mice are monitored throughout the experiments. Mouse body weight before and after JEV-MEP vaccination needs to be monitored and shown.

=> Details of animal experiments, including gender and age of mice, have already been specified in M&M (section 4.6). However, we also think that it would be helpful to describe the explanation in the Figure legend for readers' understanding. According to your kind suggestion, the details of the animal experiment were additionally described in the legend of Fig. 4a.

In these experiments we do not have results of mouse weight measurements before and after vaccination. However, external phenotype changes were periodically monitored for the mice used in the experiment, and anorexia, dullness, and hunchback were not observed following vaccination. The corresponding monitoring results were additionally described in the text (lines 175-177).

5)    Figure 4b, at least 3 mice need to be included in control / PBS group for statistical analysis. In addition to showing the results from each individual mouse, summarized data with mean and standard deviation needs to be plotted between PBS and JEV-MEP groups. The results need to be compared using statistical analysis and p-value needs to be presented.

=> Thanks for the advice on critical points. We collected the experimental results performed under the same experimental conditions as the experiments reflected in the existing figure, newly derived the experimental results with increased N number, and conducted statistical analysis on them.

In addition, according to the advice, the previous graph was replaced with a graph that can provide information on the average and standard deviation of each group, and the existing graph of individual mouse results was inserted in supplementary figure 5a.

6)    Figure 4c, the raw data calculated for PRNT titer needs to be shown and statistical analysis with mean and standard deviation needs to be presented.

=> Unfortunately, the results of our PRNT titre measurement are not able to process statistically due to the lack of N number. Due to the nature of the PRNT test, which requires the use of an infectious virus and appropriate facilities, it is difficult to perform additional experiments to secure the N number. Nevertheless, the experimental results show a clear neutralizing effect by JEV-MEP vaccinated mice. To aid readers' understanding of the results, raw data and calculation formulas for the PRNT test were added to supplementary Figure 5b.

7)    Figure 4, the positive control group using standard vaccine material to immunize mice is missing.

 => We could not use positive control vaccines (commercialized Japanese encephalitis vaccine products, etc.) in the vaccine efficacy evaluation experiment due to material supply problems. However, in the ELISA experiment, using the positive sample (anti-JEV E antibody) enclosed in the kit, it was confirmed that the serum of mice inoculated with our JEV-MEP had a titre comparable to the antibody. The measured value of the Anti JEV envelop protein E antiserum used as a positive control in the ELISA experiment of Fig. 4b is additionally shown in revised version.

The safety concerns of using the plant-based expression system of JEV-MEP as vaccine against JEV need to be discussed.

=> We added an explanation about the safety concerns of plant-based expression systems in the discussion.

Reviewer 2 Report

The authors have described the purification of the JEV-MEP antigen 83 from N. benthamiana, as well as the description of its ability to induce an immune response in mice.

Major issues

4.1. The sequences used must be described in detail or a reference must be provided.

4.6. Were confirmed positive and confirmed negative mice used?

4.7. Can you please clarify what was used as positive control and negative control sera?

Please describe how the data were managed and analysed.

Minor issues

Figure 2. Further depictions of the findings can be added as supplementary material.

2.3. Some of the descriptions should be fit better as M & M.

Figure 3. Also for this figure, further depictions of the findings can be added as supplementary material.

2.5. Again, some of the descriptions should be fit better as M & M.

Tabulation of findings will help in the flow of reading and understanding of the results.

Discussion. Please divide in two sub-sections.

Moderate editing of English language required.

Author Response

To reviewer,

We would like to thank you for your review and corrections. Regarding major and minor points, we corrected the manuscript and explained it here in red color. 

Major issues

4.1. The sequences used must be described in detail or a reference must be provided.

=> Amino sequence of pSK480, and descriptions are now added as Supplementary Figure 2.

4.6. Were confirmed positive and confirmed negative mice used?

4.7. Can you please clarify what was used as positive control and negative control sera?

4.6 & 4.7 => We used PBS-administered mouse serum as a negative control in the JEV-MEP vaccine efficacy animal experiment. As positive control, unfortunately, we could not use positive control vaccines (commercialized Japanese encephalitis vaccine products, etc.) due to material supply problems. Instead, in the ELISA experiment, using the positive sample (Anti JEV envelop protein E antiserum) enclosed in the kit, it was confirmed that the serum of mice inoculated with our JEV-MEP had a titre comparable to the antibody. The measured value of the JEV E antibody used as a positive control in the ELISA experiment of Fig. 4b is additionally shown in revised version.

Please describe how the data were managed and analysed.

=>Section 4.8 is now added to describe how quantification and statistical analysis were conducted.

Minor issues

Figure 2. Further depictions of the findings can be added as supplementary material.

=> We think Figure 2 contains enough data for describing optimal conditions for the production of JEV-MEP. But we would supplement it if you let us know what kinds of further depictions would improve this manuscript.

2.3. Some of the descriptions should be fit better as M & M.

=> Some of descriptions in the sections 2.3 and 4.3 were corrected.

Figure 3. Also for this figure, further depictions of the findings can be added as supplementary material.

=> We think supplementary Figure 2 (Supplementary Figure 3 in this revised version) already explained the purification processes and details. But we would supplement it if you suggest any other information that is supposed to be added.

2.5. Again, some of the descriptions should be fit better as M & M.

=>We think descriptions in 2.5 is fit to results and important for the reader’s understanding.

Tabulation of findings will help in the flow of reading and understanding of the results.

=> We think Supplementary Figure 1 describes the workflow of this study for better understanding.

Discussion. Please divide in two sub-sections.

=>Discussion is now divided into two subsections.

Round 2

Reviewer 2 Report

The manuscript has been greatly improved.
In relation to further visualization, whilst I understand the point of the authors, for the benefit of future readers, I would favour the inclusion of more depictions as supplementary material.

Author Response

Thank you for reviewing our manuscript. To the reviewer’s comment, we prepared answers to the comments in red color.

Comments and Suggestions for Authors

The manuscript has been greatly improved. In relation to further visualization, whilst I understand the point of the authors, for the benefit of future readers, I would favour the inclusion of more depictions as supplementary material.
=>Thank you for your help with the improvement of the manuscript.
We don’t think we have any more data for supplementary materials. But the Legends of Supplementary Figures were corrected and supplemented with more depictions for future leader’s better understanding.